# The Farm's Orientation towards Sustainability: An Assessment Using FADN Data in Italy

Concetta Cardillo *, Antonella Di Fonzo and Claudio Liberati

Council for Agricultural Research and Economics—Research Centre for Agricultural Policies and Bioeconomy, Via Barberini, 36, 00187 Rome, Italy
* Correspondence: concetta.cardillo@crea.gov.it

**Abstract:** The new Common Agricultural Policy (CAP) for the period 2023–2027 commits farmers towards achieving ambitious environmental objectives through farm organization and management. This European agricultural policy has adapted to the contemporary challenges faced by the new model of agricultural development. It aims to enhance the contribution of agriculture to the EU's environmental and climate objectives while providing better targeted support to small-scale farmers to promote farms' competitiveness. The main objective of this paper is to describe Italian farms and classify them into groups based on their main characteristics, as well as to analyze their performance and behavior in terms of sustainability and competitiveness. The novelty and innovativeness of this study are found in the data used; a 2020 dataset from the Italian Farm Accounting Data Network (FADN) was used. The quality of FADN data in farm sustainability assessment is widely acknowledge in the literature. To achieve the purpose of this study, a multivariate analysis, in particular, the Principal Components Analysis (PCA), and a Cluster Analysis (CA) were applied. These analyses helped us to obtain the factorial axes which then enabled us to identify economic information on farms, and a better interpretation of farmers' aptitude to undertake environmentally friendly actions. As a result, eight groups of farms were identified, and their characteristics and performance were described at Italian district level. The results of the study reflect the influence of European interventions towards encouraging farmers to use more environmentally friendly agricultural practices. Under this perspective, the findings contribute to the current debate on green architecture pursued by the future European agricultural policy.

**Keywords:** farm sustainability; Farm Accounting Data Network; cluster analysis; multivariate analysis





## 1. Introduction

In recent years, increased consumer awareness of the negative environmental impact of agro-industrial production has prompted farms in the primary sector to use increasingly sustainable production models. Consumers are concerned about the high volume of food waste related to the food industry and its consequences on the environment. The increasing consumer awareness and concern is largely attributed to the advent of green technologies. That is, the emergence of new green technologies provides alternative and sustainable ways of producing food which consumer groups yearn for. This calls for sustainable models of production and consumption [1]. From a multifunctional agriculture perspective, farm competitiveness and sustainable practices cannot be disentangled if the goal of a policy is to minimize the negative environmental impact of food production [2]. This is consistent with the specific objectives of the 2014–2020 CAP. The policy provides new development opportunities linked to increasing consumer interest in sustainable products. In 2017, the European Commission presented a communication to the European Parliament [3] containing the new legislative proposals [4] for the reform of the CAP 2021–2027. The Commission included three general objectives in its strategy, including "setting higher ambitions for action in favor of the environment and climate", and nine strategic objectives

focusing on social, environmental, and economic aspects [4]. Currently, the post-2020 CAP reform aims to reconcile economic growth and environmental protection goals from a sustainable perspective. Hence, the future CAP will address new challenges related to climate change and sustainability. It will also focus on increasing competitiveness through interventions that favor generational renewal, knowledge transfer, and access of young farmers to land, thus redeveloping the social role of farmers. The new European agricultural policy aims to support sustainable production to enhance the agricultural productive fabric in terms of competitiveness [3,4]. Although the post-2020 CAP includes the implementation of actions that enhance the sustainable and competitive dimension of agriculture in its multifunctional dimension, the need arises to investigate the effectiveness of rural development interventions that focus on promoting competitiveness, environment sustainability, and poorly resource-endowed areas. From a continuous legislative evolution perspective, this article aims to provide a description of the productive fabric of Italian farms and evaluate farm positioning in terms of sustainability and competitiveness. As highlighted by some research studies, the transition to a green agricultural model represents a strategic lever to guarantee the resilience and competitiveness of agricultural farms [5,6]. Consequently, studies that aim to contribute to the debate on farm competitiveness, resilience, and sustainability ought to use data that adequately cover the dimensions of the issues. In this regard, the European Commission has started the process of converting the Farm Accounting Data Network (FADN) into a sustainability network—that is, the Farm Sustainability Data Network (FSDN)—by evaluating a list of possible new environmental and social variables. The goal of this evaluation is to identify whether new variables are present in explaining farm competitiveness, sustainability, and resilience. Using the FADN data set, this paper aims to offer a contribution to the ongoing debate on productivity and environmental performance in agriculture. The efficacy of the FADN database in assessing agricultural intervention is well documented in the scientific literature [7]. Our study provides evidence on the efficacy of FADN data to assess farm-level sustainability and competitiveness. In this way, the study adds knowledge to the literature on farm-level sustainability and competitiveness assessment. Data on structural characteristics, attitudes, and environmentally friendly behaviors of farmers are analyzed using factor analysis in the specification of the principal components analysis. The farms are then grouped into homogeneous groups defined through a cluster analysis for a clear explanation of profiles. The final considerations of this study are intended to represent a contribution to the current challenge pursued by the common agricultural policy to ensure a smooth transition to the new CAP reforms, describing the important support that the FADN provides in assessment of the effectiveness of agricultural interventions to achieve virtuous levels of competitiveness and sustainability. The document is organized into five sections: Section 1 provides an introduction of the study; Section 2 is dedicated to a literature review and argues on the importance of FADN data in competitiveness and environmental sustainability assessment; Section 3 describes the data used and discusses the methodology; Section 4 provides discussions and findings; and Section 5 concludes with recommendations for future studies.

## 2. Literature Review: The Use of FADN Data in Agricultural Intervention Assessment

In the last decade, the affirmation of globalization both in consumption and in production has pushed the EU to promote increasingly sustainable interventions and actions within the EU single market. The sustainable agriculture model has gained relevance since the publication of the Brundtland Report in 1987 [8].

International attention to the environment, global ecological disasters, and the spread of the idea of Sustainable Development birthed the Brundtland Report in 1987 [8]. In the following years, numerous conferences on sustainability broadened the definition of sustainability. For instance, the World Summit on Sustainable Development in Johannesburg in 2002 interpreted sustainability as a dynamic concept, valid both at the level of decision-making and farming. This suggests that sustainability can be defined as the

indivisibility of three components, that is, a development model that combines economic, environmental, and social sustainability. The new Common Agricultural Policy 2023–2027 strongly emphasizes environmental sustainability and enhancing the competitiveness of agriculture and rural areas. In fact, in the past programming period of 2014–2020, the improvement of competitiveness and the environment in the agricultural sector was envisaged. Consequently, the new legislation due to begin in 2023 contains strategic objectives to promote competitiveness and environmental sustainability. To reach the goal of a competitive and sustainable agricultural sector, each Member State is required to adopt its own National Strategic Plan drawn up and approved by the European Commission. This is meant to be an indication of a unitary strategy for implementing the various financial instruments available for enhancing the competitiveness of the agro-food system from a sustainable perspective. The plan is specific for each country, as there are different factors, both endogenous and exogenous, that influence the economic performance and competitiveness of farms [9]. Under this legislation, the concept of multifunctional agriculture is consolidated, as well as the issues associated with it, such as environmental protection and biodiversity. The new model of implementation of the CAP is based on the effectiveness of rigorous policy interventions in contributing to the EU's environmental and climate objectives in agriculture. A current study illustrates how relevant it is to study the impact of the common agricultural policy on the level of economic sustainability of farms in the European Union [10]. In this context, there is a need for statistical information, including data on economic, environmental, and social practices. In line with the CAP objectives, these data are used to implement and evaluate the impacts of evidence-based policy to improve the assessment of farm sustainability. However, considering the new CAP reform, EU programs need to be assessed at the European level. The request for the assessment of European interventions in agriculture increases the need to have statistics capable of combining the environmental dimension with economic performance in agriculture. In particular, the need arises to investigate the impact of CAP interventions in agriculture on farms to encourage farmers to use more environmentally friendly agricultural practices. At the same time, a better connection of support to farm results and performance is required. Regarding this issue, one of the most discussed topics in the literature is the dynamism of the concept of sustainability and its measurement process. This discussion largely rests on the scarcity of quality data. Research studies highlight that sustainability is a multidimensional concept and attempts to measure sustainability in an established information framework may not guarantee measurement accuracy [11]. In Europe, there are several data sources that meet the information needs of agricultural data such as the FADN. At the farm level, the existing FADN data refer to the aspects that best express the technical and economic efficiency of farms [12], which are not fully focused on the issues of green architecture that are driving the new CAP. Environmental issues play a key role when it comes to policy assessment. In the context of the assessment of agricultural sustainability, Figuières et al. [13] suggest considering not only the farm-level characteristics, but also the existing interdependence between farms and the economic–productive context in which they operate. To this end, scientific studies show that the FADN represents an important statistical tool when it comes to evaluating European interventions in agriculture from a competitiveness and sustainability perspective. As suggested by Hennessy and Kinsella [14], a key strength of the FADN database is that it is a source of statistical information on a wide range of farms, thus facilitating comparative analysis among different groups of farms.

The scientific literature suggests the relevant support that FADN data can provide regarding sustainability assessment and monitoring results among farms [15,16], and that it is an important information source for understanding the impact of the CAP measures [17]. Monitoring of results constitutes one of the strategies that strengthens and influences the adaptability of farms [18], without which it is not possible to hypothesize sustainability in its three dimensions. On a similar trajectory, Poppe and Vrolijk [19] investigated existing methods to collect data on the sustainability of farms. The authors, through the publication

of the results of the FLINT project, reported the appropriateness of FADN data in detecting sustainability dimensions. They showed that FADN data adequately express the heterogeneity of the agricultural sector in the EU. Their research findings provide grounds to widen the FADN database with appropriate data to explain the multidisciplinary nature linked to the sustainability components. In the same context, Buckley et al. [20] reported the possibility of expanding the FADN to derive indicators of nitrogen efficiency in the dairy sector. Despite the consolidated idea that the FADN can respond to the information needs expressed by the new CAP, the literature expresses interest in the availability of punctual surveys and a systematic collection of information about the environment. Kelly et al. [7] showed that there is a lot of research in the literature that highlights the suitability of FADN data for studies of sustainability and competitiveness in agriculture. The authors identified research that uses data exclusively from the FADN [21–25], and research using FADN data in combination with national initiatives that collect additional data through the FADN [26–30]. Finally, they also detected studies and research using FADN data in combination with additional data from sources other than FADN, available at national, EU, or international level [31–34].

## 3. Materials and Methods

### 3.1. Description of the Information and Database Used

To achieve the purpose of the analysis proposed, we used data collected by the Italian Farm Accountancy Data Network (FADN). These data are obtained through a sample survey carried out every year by each member state of the European Union, according to a common methodology. The survey was established in 1965 with EEC regulation number 79/1965, and subsequently revised by several regulations (more recent is Reg. EU n. 1652/2020). All the regulations aimed to provide information to measure the evolution of the incomes of agricultural entrepreneurs and the functioning of farms.

The FADN field of observation is represented by all the farms that achieve a certain threshold of Standard Output (SO) (for Italy, this threshold is now EUR 8000), and the information base used for the extraction of the sample is the Agricultural Census, updated through the Farm Structures Surveys (FSS). The stratification adopted in the sample design is carried out according to three main dimensions: the geographical region, economic size, and Type of Farming (ToF).

Although FADN was created as a survey tool mainly oriented to the economic, financial, and structural aspects of the farm, in recent years the objectives pursued have diversified and expanded, given the considerable availability of information related to structural, accounting, and non-accounting nature. The great amount of available information allows us to achieve new purposes such as monitoring the evolution of agricultural income and evaluating the impact of European or regional agricultural policies, or the environmental impact of agricultural businesses. Therefore, the FADN database represents the only harmonized source of structural, economic, social, and environmental data on agricultural holdings, which covers the entire EU and thus allows a comparative analysis at European level. (For more information about FADN methodology please consult documents available at https://agridata.ec.europa.eu/extensions/FarmEconomyFocus/FarmEconomyFocus.html (accessed on 10 September 2022) and https://agriculture.ec.europa.eu/data-and-analysis/farm-structures-and-economics/fadn_en (accessed on 10 September 2022); For more information about Italian FADN please check at https://rica.crea.gov.it/ (accessed on 22 August 2022).

In each member state of the EU, there is an official liaison agency that coordinates the collection and processing of FADN. In Italy, this agency is represented by the Council for Research in Agriculture and Agricultural Economic Analysis (CREA). The survey is realized through a network of data collectors who conduct face to face interviews with farmers, utilizing a questionnaire that includes information on structures, production, and economic results, as well as a range of information related to social and environmental aspects.

The field of observation of our study consists of the Italian FADN sample referring to 2020. It is the latest dataset available, and it contains data on 10,764 farms divided by geographical districts. The choice to analyze the phenomenon by geographical distribution (Figure 1) lies in the configuration of the topic as site-specific. In particular, the national productive fabric seems to be characterized by a marked structural dynamic and by the presence of a multifunctional agriculture model with an approach to the diversification of activities of not-negligible importance. The emergence of these dimensions has made it possible to compare, in terms of competitiveness and sustainability, the main effects of the last programming period and the one currently underway. The final time reference allows us to interpret the results of the processing as a description of the status quo with respect to the implementation of the 2014–2020 regional rural development plan. Table 1 shows the main characteristics of farms in the sample utilized, in terms of number of farms, hectares of Total Agricultural Area (TAA), and Utilized Agricultural Area (UAA).

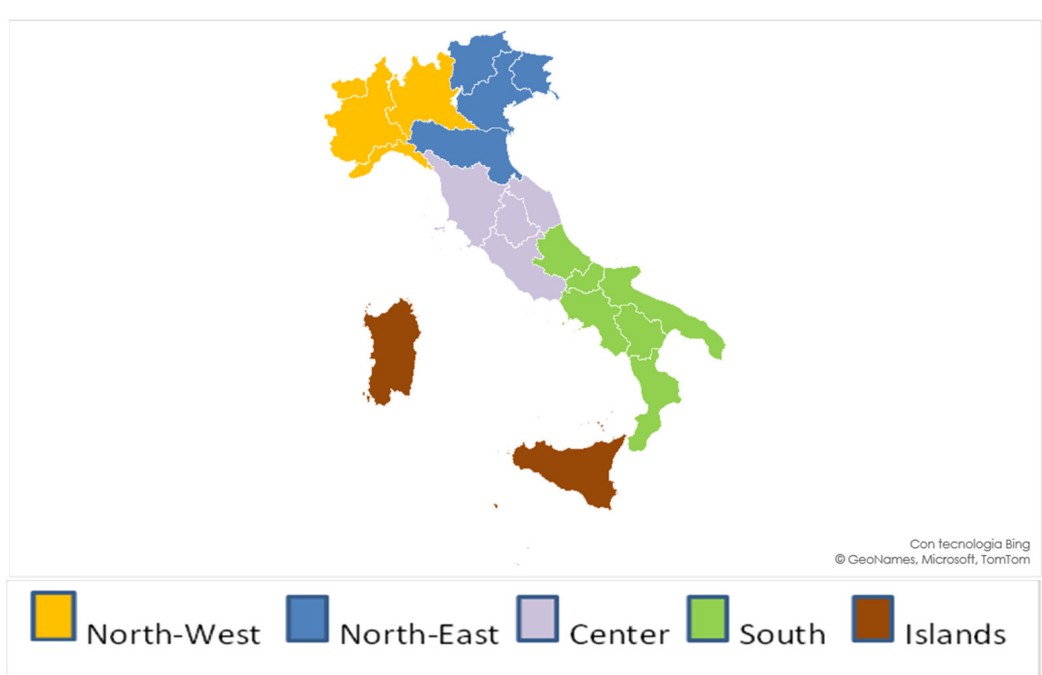

**Figure 1.** Map of Italian geographical districts.

**Table 1.** Main characteristics of FADN farms in Italian geographical districts.

| District | Number of Farms | TAA (ha) | Average TAA (ha) | UAA (ha) | Average UAA (ha) |
|---|---|---|---|---|---|
| North-East (NE) | 2524 | 88,347.23 | 35 | 71,513.78 | 28.3 |
| North-West (NW) | 1951 | 90,932.16 | 46.6 | 74,460.54 | 38.16 |
| Centre (CEN) | 2011 | 92,864.7 | 46.17 | 76,397.95 | 38 |
| South (MER) | 3075 | 90,671.98 | 29.48 | 83,645.36 | 27.2 |
| Islands (INS) | 1201 | 58,427.53 | 48.64 | 54,773.24 | 45.6 |

*3.2. The Methodology of Cluster Analysis and Principal Component Analysis*

The methodology used to transform the set of observed variables into synthetic indicators of the strategic approach follows a consolidated approach. The statistical methodologies used fall within the multivariate analysis techniques, specifically, Principal Component Analysis (PCA) and Cluster Analysis (CA). Principal component analysis aims to observe and identify the existing relationships between a set of qualitative variables observed on a group of statistical units. The cluster analysis method makes it possible to subdivide the set of statistical units under study into entirely homogeneous subsets according to the metric chosen for calculating the distances in the dimensional space generated by the n observed variables with a minimum loss of information [35].

Sabbatini (2011) applied a consolidated study methodology, called typological analysis, to investigate the Italian agricultural context [36]. The technique uses census data to define the strategic profile of farms by means of an ex-post classification based on the sequential application of an analysis of principal components and a cluster analysis. The subsequent aggregation of the profiles defined at the farm level allows the regional and territorial agricultural context to be outlined. Our paper provides an exemplification of the model using FADN data and PCA and Factor Analysis (FA), which are techniques aimed at reducing the dimensionality of a set of data for exploratory data visualization, and for a better interpretation of the strategic profiles of the farms that emerge and from which the interpretation of the factorial axes derives. As we already stated, the data used in this research were obtained through a FADN database that was developed to support the processing of PCA, CA, and FA at farm level. To this end, data were analyzed using SPAD software, version 5.0, realized by CISIA-Ceresta (France). A descriptive analysis of average values was presented in table format to discuss different farm features in general terms, i.e., at sample level. The variables and index values for farm profiles were used as segmentation variables in a cluster analysis. Consequently, a hierarchical cluster analysis [37] was performed to identify farm profiles that differed according to their sustainability and competitiveness. The number of clusters was provided directly by the SPAD software which can identify the optimal aggregation of farms. In detail, parameters for the characterization of clusters were carried out by continuous modalities and frequencies, through the application of a ranging criterion of mean values and percentage of test values, respectively.

The analysis has been carried out at district level because, as we already stated, the Italian agricultural sector is characterized by a strong territorial differentiation, both in the structural aspects and in the management of agricultural system [38]. To facilitate reading and allow comparison between the different districts considered, the farms are summarized in a small number of homogeneous groups defined through a CA. This analysis led to the separate classification of farms on the basis of the indices that described the set of relationships between the farm, the environment, and competitiveness. In this way, the five typological classifications were defined, each of which represents an interaction between farm profile and external environment. The application of the PCA has made it possible to identify the factors that contribute to explaining the differentiations in the strategic profiles of farms. The method makes it possible to place each farm in a space defined by these factors, attributing a factorial score to each observation, or a numerical value obtained through a linear combination of the variables used.

Following this approach, the variables collected by the FADN survey were selected and processed to calculate 25 indices, shown in Table 2. Table 2 is useful for describing the reality of Italian farms and the factors used as active variables in the PCA.

**Table 2.** Description of the indexes used in the Principal Components Analysis *.

| Indexes | Indexes Description |
| --- | --- |
| 1. Arable crops area rate | Arable_crops area/UAA: it indicates the arable land area incidence compared to the utilized agricultural area. |
| 2. Current cost rate | Current_Cost/GSP: it indicates the current cost incidence compared to the total gross salable production. |
| 3. European subsidies rate | Sub_EU/GSP: it indicates European subsidies incidence compared to the gross salable production. |
| 4. Family labor rate | FWU/AWU: it indicates the unpaid labor incidence compared to the farm's total labor force. |
| 5. Forest area rate | Forest_area/TAA: it indicates the forest area incidence compared to the total agricultural area. |
| 6. Gross agricultural labor productivity | GSP/AWU: it indicates the unitary productivity compared to farm revenues. |
| 7. Gross agricultural land productivity | GSP/UAA: it indicates the unitary productivity of the utilized agricultural area. |
| 8. Irrigation systems rate | Irrigation_systems/UAA: it indicates the irrigation systems incidence compared to the utilized agricultural area. |
| 9. Land capitalization | Land and buildings/AWU: it explain the intensity degree of landed capital use compared to the labor total units. |
| 10. Land intensity | Land and buildings/UAA: it indicate the soil intensity degree of the landed productive factor and of the capital invested on it. |
| 11. Land intensification degree | ALU/AWU: it indicates the availability of agricultural area for work unit. |
| 12. Land mechanization degree | kW_Machine/UAA: it indicates farm mechanization degree compared to the utilized agricultural area. |
| 13. Meadows and pastures area | Meadows_pastures_area/UAA: it explains the land used incidence for the cultivation of grass or other herbaceous forage plants compared to the utilized agricultural area. |
| 14. Net land productivity | VA/UAA: it expresses the net productivity of the utilized agricultural area. |
| 15. Net land profitability | Net_Income/UAA: it explains the net profitability of family work. |
| 16. Nitrogen rate | Nitrogen_per_hectare/UAA: it indicates the amount of nitrogen used compared to the utilized agricultural area. |
| 17. Phosphorus rate | Phosphorus_per_hectare/UAA: it indicates the amount of phosphorus used compared to the utilized agricultural area. |
| 18. GSP direct sales rate | GSP_direct sales/GSP: it indicates the gross salable production incidence relating to direct sales compared to total gross salable production. |
| 19. GSP processing rate | GSP_processing/GSP: it indicates the gross salable production incidence relating to processing compared to the total gross salable production. |
| 20. GSP quality rate | GSP_quality/GSP: it indicates the gross salable production incidence relating to quality compared to the total gross salable production. |
| 21. Potassium rate | Potassium_per_hectare/UAA: it indicates the amount of potassium used compared to the utilized agricultural area. |
| 22. Tree area rate | Tree_area/UAA: it expresses the incidence relating to area destined for tree crops compared to the utilized agricultural area. |
| 23. UAA rate | UAA/TAA: it indicates the utilized agricultural area incidence compared to the total agricultural area. |
| 24. ALU rate | ALU/UAA: it indicates the livestock unit incidence compared to the utilized agricultural area. |
| 25. Water usage | Total_water_volume/UAA: it explains the water volume used compared to the utilized agricultural area. |

* GSP: Gross Salable Production; ALU: Adult Livestock Unit; TAA: Total Agricultural Area; UAA: Utilized Agricultural Area; FWU: Family Working Units; AWU: Annual Working Units; VA: Value Added.

## 4. Results and discussions

### 4.1. The results of the Factorial Analysis

Based on the positive or negative correlation of the variables used, factor analysis allows to define a series of factorial axes that order the factors according to their ability to capture the variability of the data and allow for a better interpretation of agricultural reality. Thus, a conceptual scheme is obtained which does not capture the phenomenon in its entirety but provides a good representation of it through the chosen combination of axis and variables. The interpretation of the factorial axes obtained through principal component analysis is reported below.

1. Factorial Axis 1—Competitiveness. This axis represents the dichotomy between rent and profit as an entrepreneurial objective to be maximized and is based on the contrast between Public Aid and the Land's Profitability and Productivity. Competitiveness becomes the farm's ability to offer adequate remuneration of factors through access to the market.

2. Factorial Axis 2—Functional Diversification. This axis shows the contrast between the productivity of the land and the presence of products of certified quality (typical and organic products), of transformation, and direct sales activities. As regards production, the two contradictory aspects are associated with the production of arable land and with the presence of permanent crops. Therefore, there is a contrast between a productivism approach and multifunctionality, considered as the multiplicity of functions performed by agricultural enterprises, in opposition to the specialization in the cultivating function.

3. Factorial Axis 3—Environmental pressure. This factor represents the contrast between the use of farmland for cultivation and that for forests and pastures. In this case, it is possible to find the opposition between a conservation strategy and one of exploitation of the land resource and highlights the different degrees of pressure that agricultural activity exerts on the environment and on the soil.

Below are the graphs relating to the biplots of the PCA in each geographical district (Figures 2–6). Lines in blue represent variables that actively contribute to the definition of the clusters, while illustrative variables are in red.

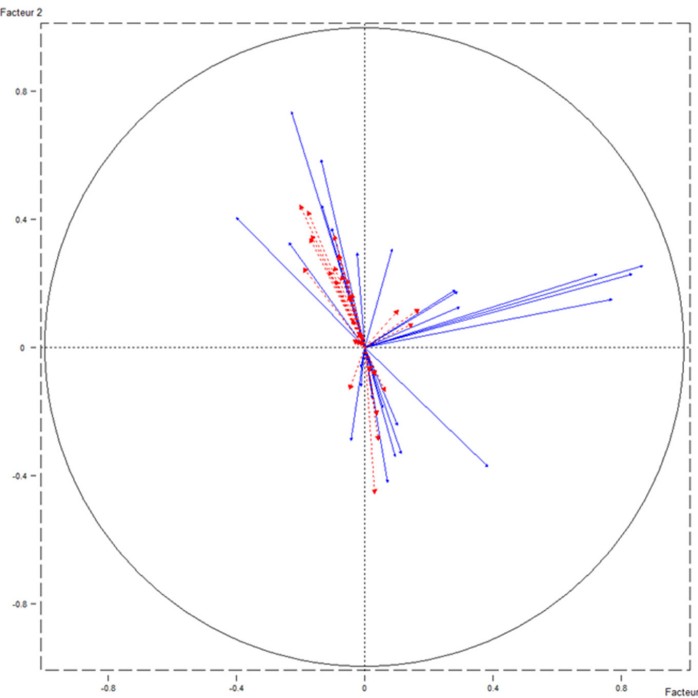

**Figure 2.** Biplot of PCA in the northwestern district.

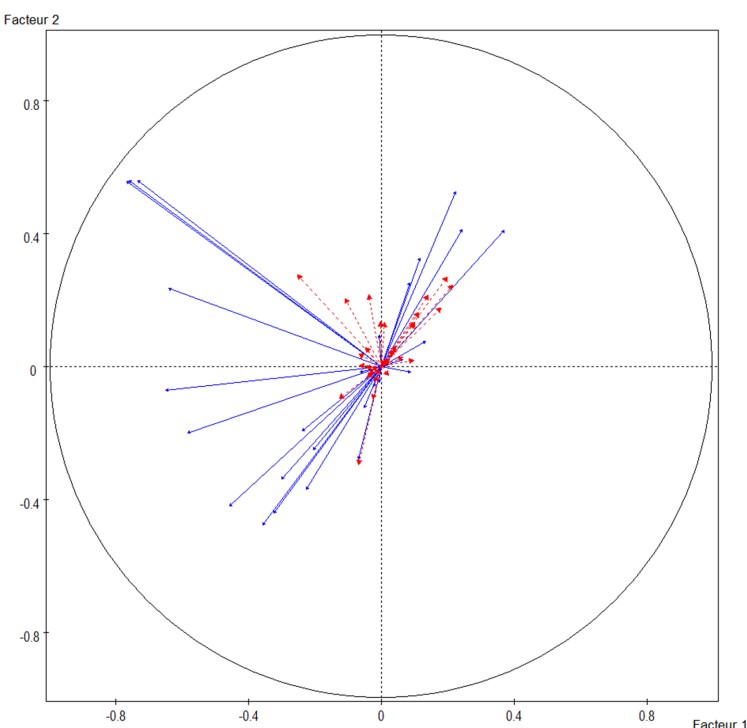

**Figure 3.** Biplot of PCA in the northeastern district.

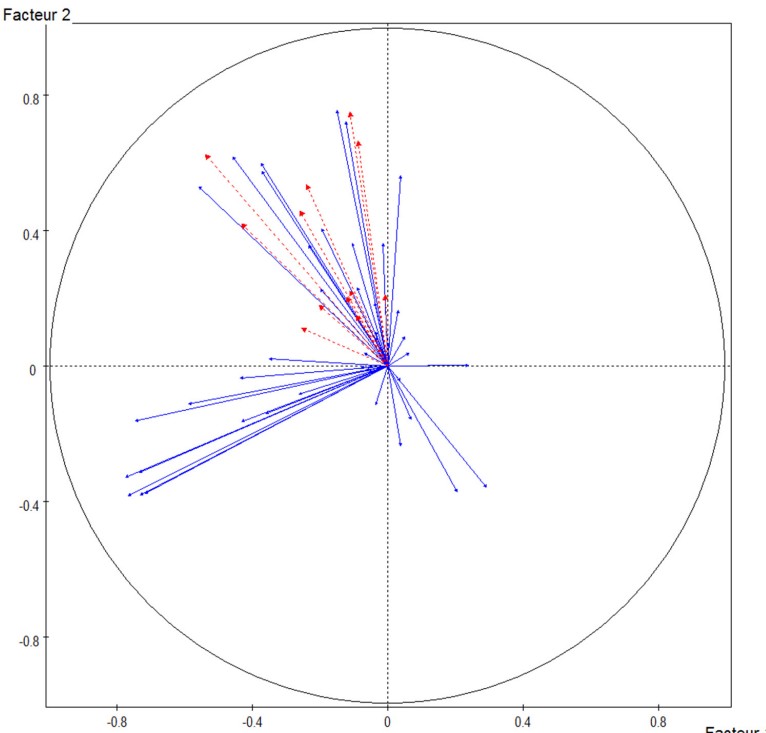

**Figure 4.** Biplot of PCA in the central district.

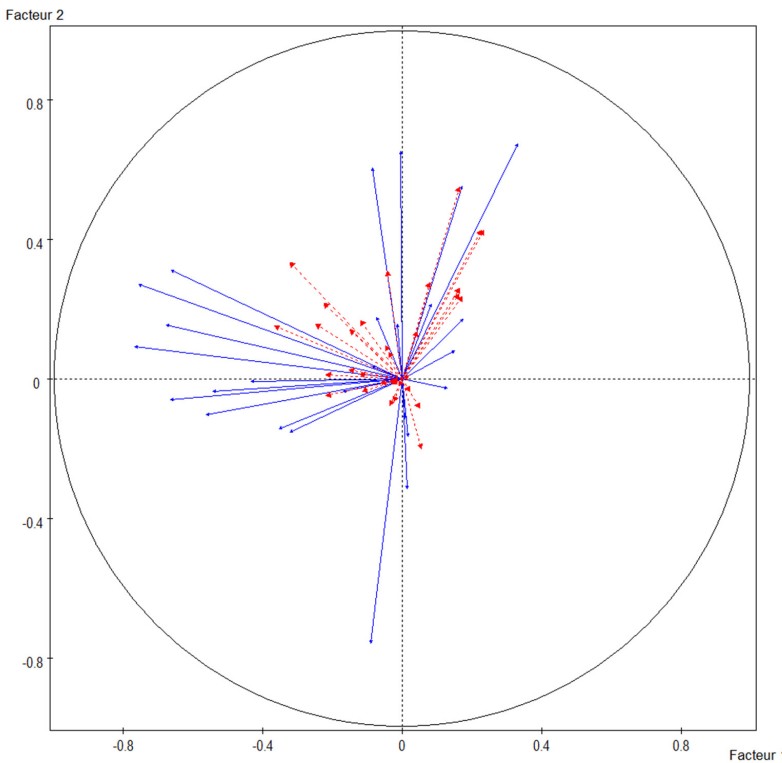

**Figure 5.** Biplot of PCA in the southern district.

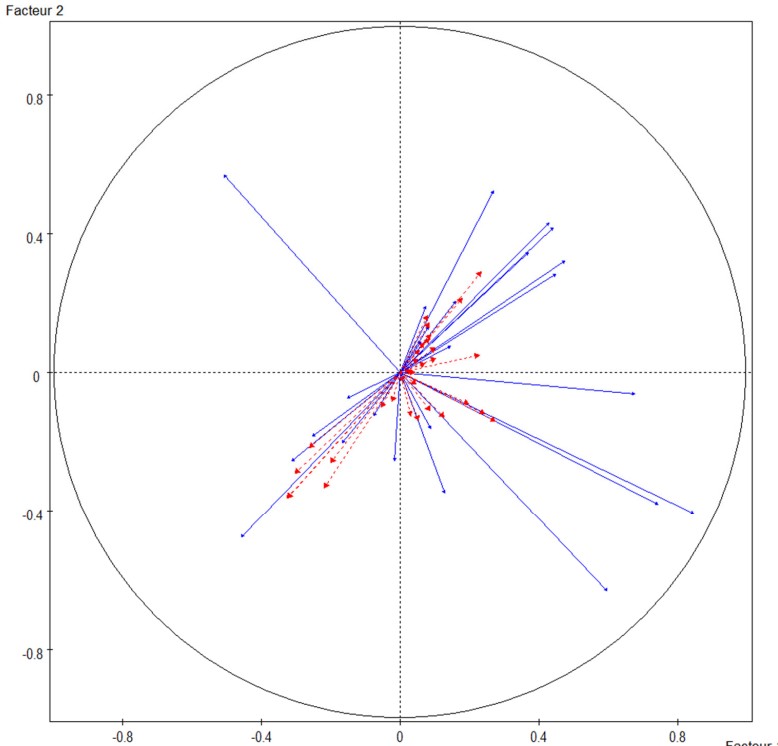

**Figure 6.** Biplot of PCA in the Islands district.

*4.2. The results of the Cluster Analysis*

The cluster analysis carried out made it possible to define eight different farm profiles (Table 3) and their description is provided below. Moreover, cluster analysis graphs and dendrograms in different geographic districts are listed in Appendix A (Figures A1–A5). Some of these profiles are present in all the districts while others are specific to one or two of the geographical divisions.

**Table 3.** Number of farms in each cluster by geographical division.

| | Homologated Family Farms | Large Capitalized Farms | Resilience | Services Farms | Short Supply Chain and Agro-Food Processing | Intensive Farm | Farm with Quality Label | Livestock Farm |
|---|---|---|---|---|---|---|---|---|
| North-West | 822 | 1123 | 3 | - | 3 | - | - | - |
| North-East | 756 | 1559 | - | 22 | 187 | - | - | - |
| Center | 365 | 1298 | - | - | 79 | - | 269 | - |
| South | 1780 | - | 16 | - | - | 22 | - | 1257 |
| Islands | 818 | 18 | - | 1 | 364 | - | - | - |

Cluster 1: Homologated family farms. This typology of farms is common to all the geographical districts representing about 42% of the entire sample of farms. A larger amount is localized in the southern regions. The farms included in this cluster are characterized by a medium-large size; the tenant and his family provide almost all the work performed (up to 80–90%). Farms also make intensive use of the agricultural area (85–90% of the TAA is represented by UAA), but the values of the profitability indices are below the regional average because the endowment of land, and mechanical and livestock capital is lower than the average regional value. In addition, European aid has a high incidence on farm income. The absence of significant diversification of farm activities and use of quality brands, associated with a large presence of family gardens, leads to the conclusion that for this kind of farm, the main function is that of residential and self-consumption. The orientation of farms is towards development paths linked to economies of scale and cultivation.

Cluster 2: Large capitalized farms. This type of farm is also numerous (37% of the FADN sample). It is present in all geographical areas apart from the south. Furthermore, the large size of this group makes it particularly important for policy evaluation purposes. Unlike the previous group, the endowment of land, and mechanical and livestock capital, is quite large and allows the achievement of high values of GSP even if the work employed is relatively limited, leading to particularly high values of the labor productivity indices (over EUR 130,000). Again, there is an intense exploitation of the land, demonstrated by a high incidence of UAA on TAA and the association between the large size and a productive orientation toward arable land allows these farms to benefit from considerable European payments, which significantly affect the farms' budgets. Future strategies of development appear oriented to the exploitation of economies of scale deriving from access to large capital. The labor factor and the farm tenant can benefit from high remunerations thanks to business investments.

Cluster 3: Resilience. This group includes very few farms located only in the northwest and mainly in the southern regions. The typology is characterized by the small size of UAA, low incomes, and labor almost exclusively supplied by the tenant and his family. Even the economic size is clearly below average, and the production is oriented mainly towards arable land, suggesting an activity of self-consumption and animal husbandry. Due to the small areas, EU payments are very low and do not reach EUR 5000, even if they represent a consistent part of the modest budget. These farms are therefore characterized by an absence of well-defined strategic development paths, and their survival appears to be linked essentially to the residential and fruition function, as well as the possibility of integrating with non-farm income (including retirement income).

Cluster 4: Services farm. This group of farms is present exclusively in the northeast area, with only one in the islands, and in any case, it represents a very small category in terms of numbers. These farms are distinguished from the others because of the relevant presence of the service component in the corporate balance sheet. The GSP of these farms is quite low; however, they can develop a considerable value-added and substantial incomes thanks to non-agricultural activities. Land productivity and services represent their principal strategic orientation.

Cluster 5: Short supply chain and agro-food processing. This group of farms, although they do not represent a significant share of the total sample, is particularly concentrated in the islands and in the northeast, while totally absent in the south of the country. Farms of this type are characterized by the combination of direct sales and product transformation. The main difference with respect to the previous type lies in the scarce use of quality brands. The enhancement of the product appears to be linked more to individual reputational aspects, which can also be used in the local circuit. Production is concentrated on tree crops and arable land, but the modest factor endowment limits the average PLV to values well below the regional reference. The value of EU payments also appears relatively low and its impact on the farm budget is limited. The typology is positioned towards development paths of multifunctionality and cultivation.

Cluster 6: Intensive farms. This typology is characterized by a very high productivity of the land, linked to the presence of greenhouses and nurseries. Despite the small farm size (on average of just under one hectare), the GSP reached average values of EUR 160,000 and over EUR 200,000. However, these are a very small number of farms and are present only in the south of the country.

Cluster 7: Farm with quality label. This group represents only 2.5% of the sample and is located exclusively in the central area. The main characteristic of this typology is the centrality of organic or typical brand productions within the farm mechanisms of value production. Obviously, this does not imply that the set of farms that use quality certifications are limited to this type. However, these farms see a substantial coincidence between the business activity and the use of the brand. The average incidence of branded and transformed productions on the GSP is particularly high. There is a strong orientation towards direct sales while the endowment of land, livestock, and mechanical capital appears below average. The production orientation is linked to tree crops and there is a wooded area above the regional average. The small area and the type of production limits the possibility of farms to benefit from EU payments, which in fact have a limited impact on the farm budget. The development paths associated with this typology are multifunctionality and cultivation.

Cluster 8: Livestock farms. The group is quite numerous but located exclusively in the southern area. The grazing areas in these farms represent an average of 57% of the TAA, which accounts on average for 83.33 ha. Other characteristics of the farms are substantial livestock capital (about 42 LU) and a LU/HA ratio of 0.67. The typology is associated with a strategic positioning defined by the binomial supervision–extensification.

Furthermore, the average values of the most relevant variables and structural indices of the identified clusters are presented in the tables in Appendix B for each geographical division (Tables A1–A5).

## 5. Conclusions

This article aims to evaluate the possible development strategies that direct the choices of farms towards an increasingly competitive and sustainable production model. For this purpose, a multivariate analysis (clustering) of a comparative type was carried out on the data collected from the Italian FADN survey in 2020. The farms were therefore grouped into classes based on explanatory indices of their structural characteristics and sustainable behaviors. In this way, four groups of farms have been identified for each geographical division considered, based on similar elements of competitiveness and sustainability in a set of homogeneous structural, cyclical, and environmental variables. The groups of farms emerging from the analysis do not have different characteristics but show similarities in their composition. Although this result represents the picture of a regional productive and agricultural fabric characterized by profound structural and conjunctural changes, the variation in the structural composition of the groups leads to a series of reflections. In fact, in 2020 the presence of the capitalized farms on the national territory highlights a development path of the farms aimed at a more sustainable use of production methods, such as the increase in livestock farms, whose co-presence of diversification of activities (such as the incidence of wooded and pasture areas) and significant public support reflect forms of environmental protection. On the other hand, the presence of homologated family farms and resilient farms shows the dependence of the competitive capacity of the Italian agricultural enterprise on the European payments. This is presumably due to delays, in some regions, in the use of the resources coming from the European structural funds established by the 2014–2020 RDP, which are not yet used by producers and from which farms have benefited partially. Consequently, the EU and national current objective is to accelerate the implementation of spending programs through the new transition rules. The existence of the service farms cluster, albeit slight, represents an expansion of farm functions, which reflects a multifunctional agricultural development trajectory. On the topic addressed, future research could branch in different directions. To address one of the main limitations of the current study, it would be appropriate to analyze the profiles of the selected farms, defining them in more detail for greater external validity of the results. This could be achieved, for example, through the selection of more explanatory variables of sustainability, to adequately deal with multidimensionality of the concept of sustainability. Considering the debate on the conversion to a Farm Sustainability Data Network (FSDN), future studies should consider extending the existing data collection on EU farms to data on their environmental and social practices. In this way, results will be more in line with the objectives of the new CAP. Finally, the results of our study could be applied at a regional level, or overall, at a national level to have a more complete view on the subject.

**Author Contributions:** Conceptualization, A.D.F., C.C. and C.L.; methodology, A.D.F. and C.C.; formal analysis A.D.F. and C.C.; investigation A.D.F. and C.L.; data curation, C.C.; writing—original draft preparation A.D.F.; writing—review and editing, C.L. and C.C.; supervision, C.L. All authors have read and agreed to the published version of the manuscript.

**Funding:** This research received no external funding.

**Data Availability Statement:** Not applicable.

**Acknowledgments:** The authors would like to thank Edward Kyei Twum for his willingness and for his valuable support.

**Conflicts of Interest:** The authors declare no conflict of interest.

## Appendix A

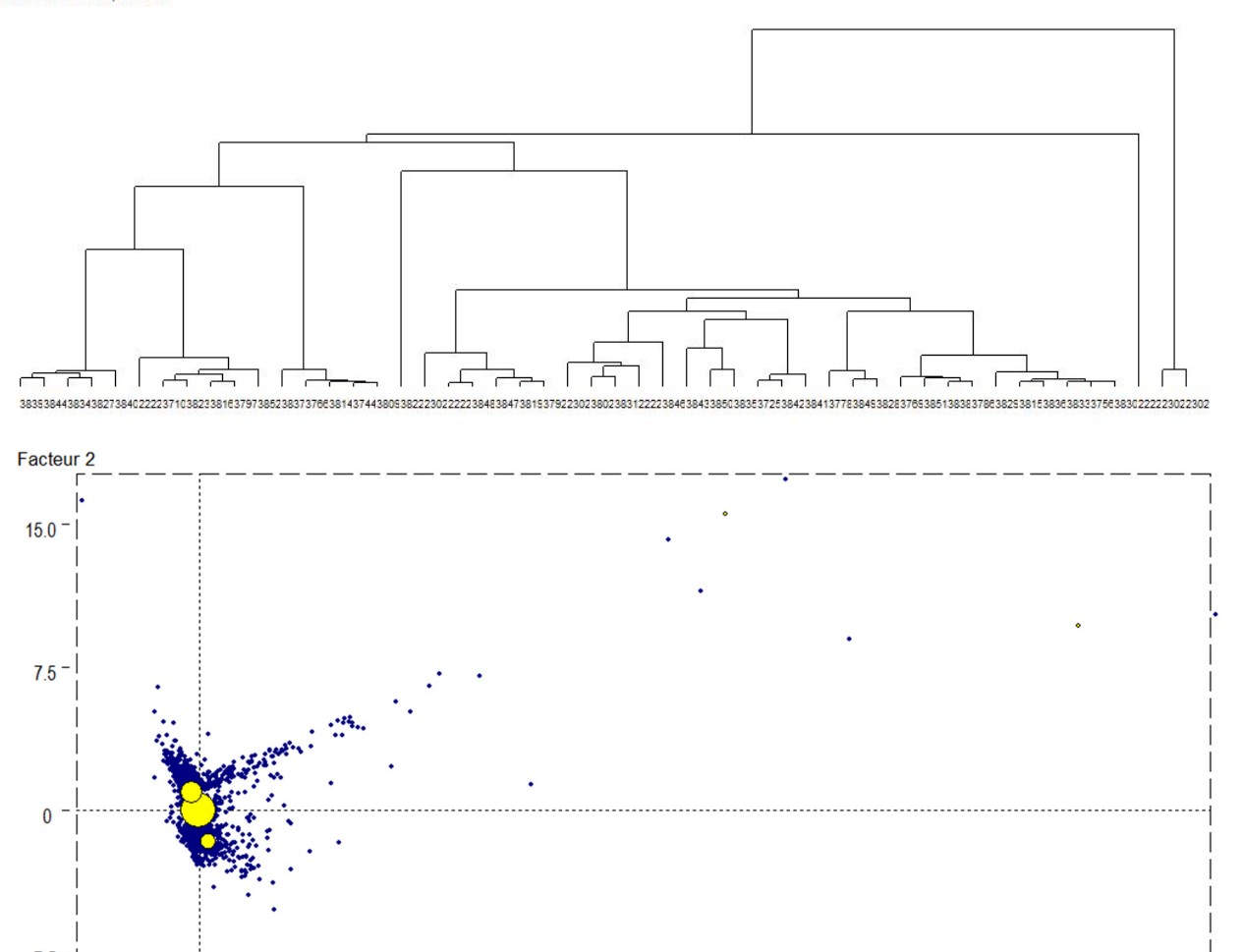

**Figure A1.** Cluster analysis dendrogram and graph for the northwest geographic distribution.

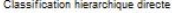

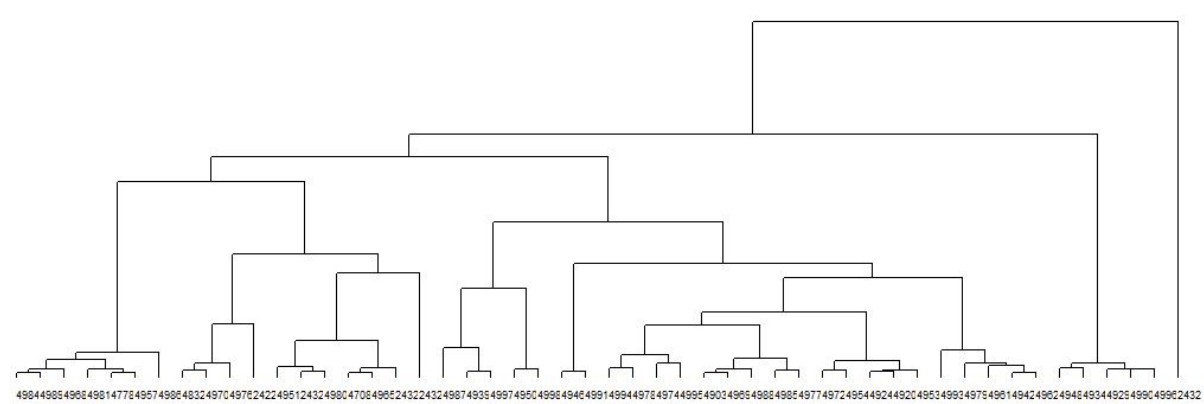

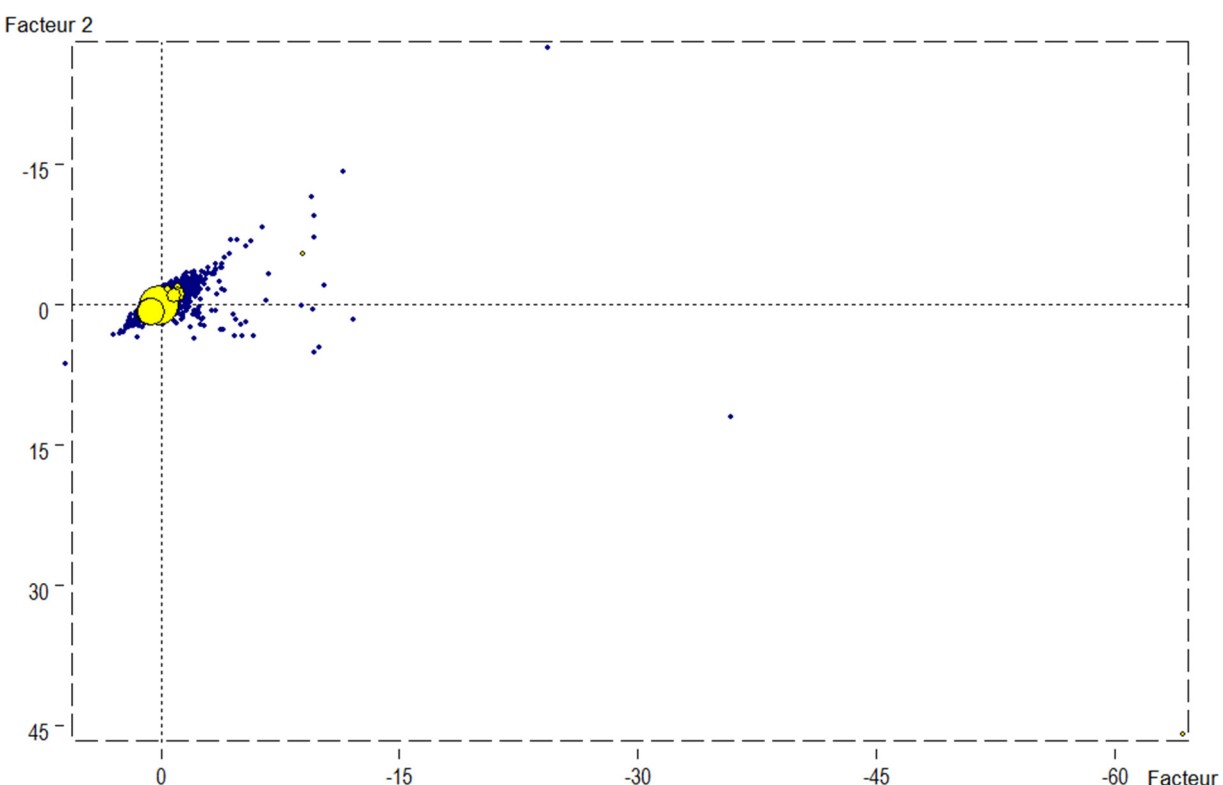

**Figure A2.** Cluster analysis dendrogram and graph for the northeast geographic distribution.

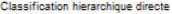

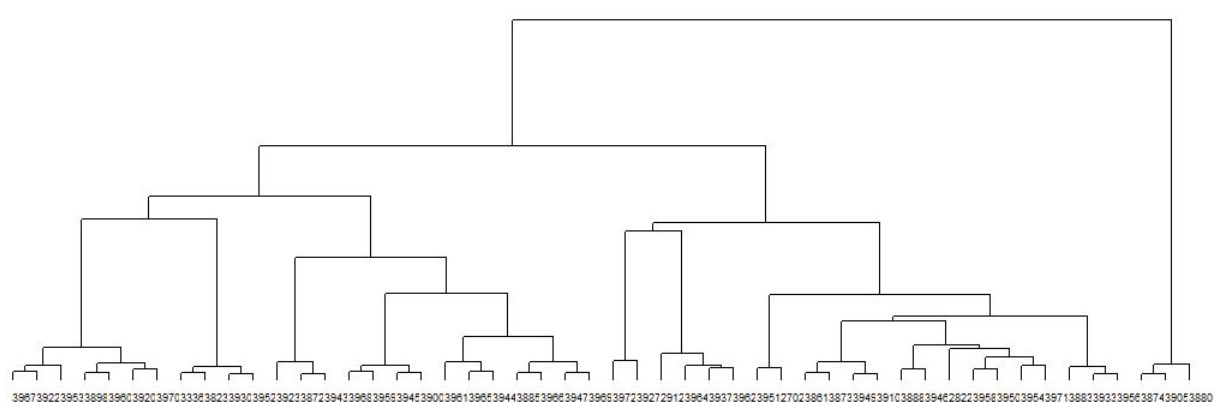

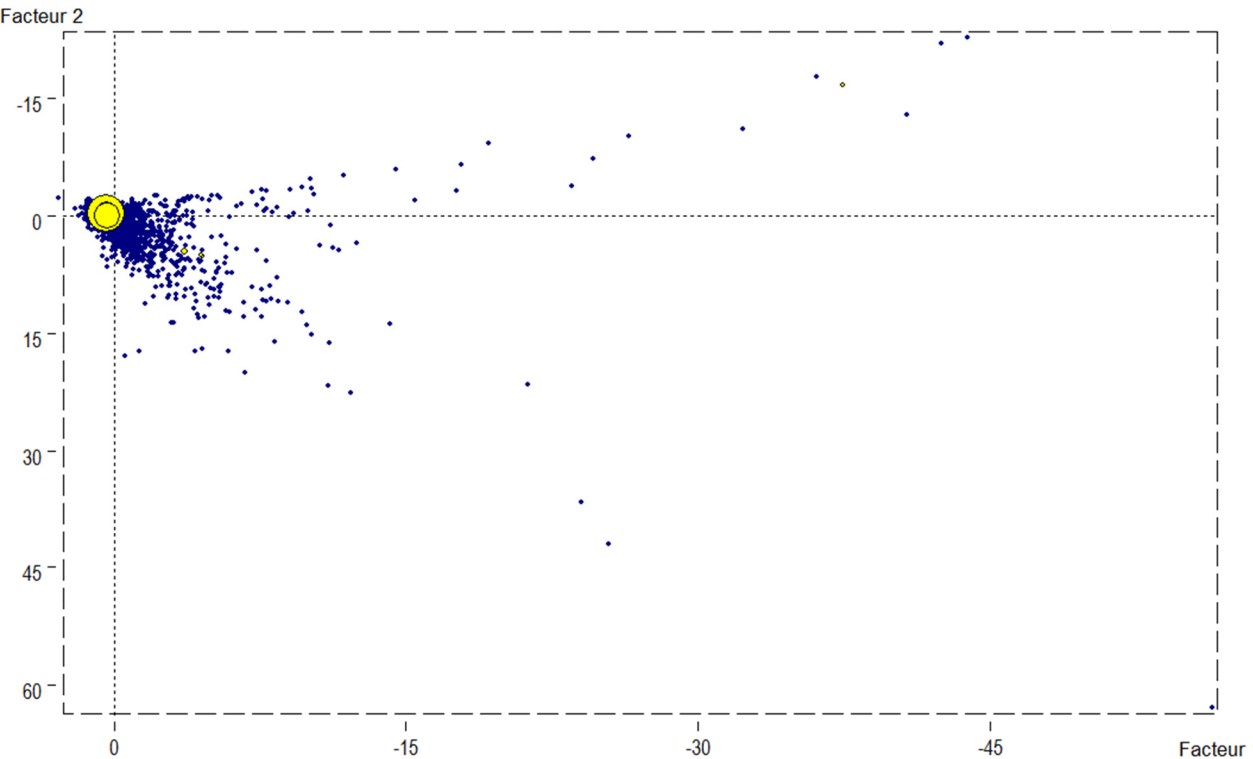

**Figure A3.** Cluster analysis dendrogram and graph for the central geographic distribution.

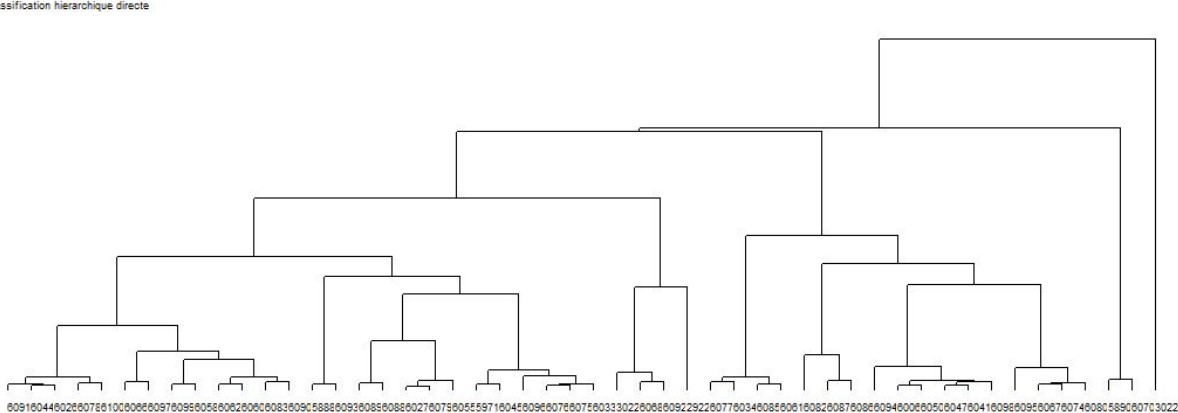

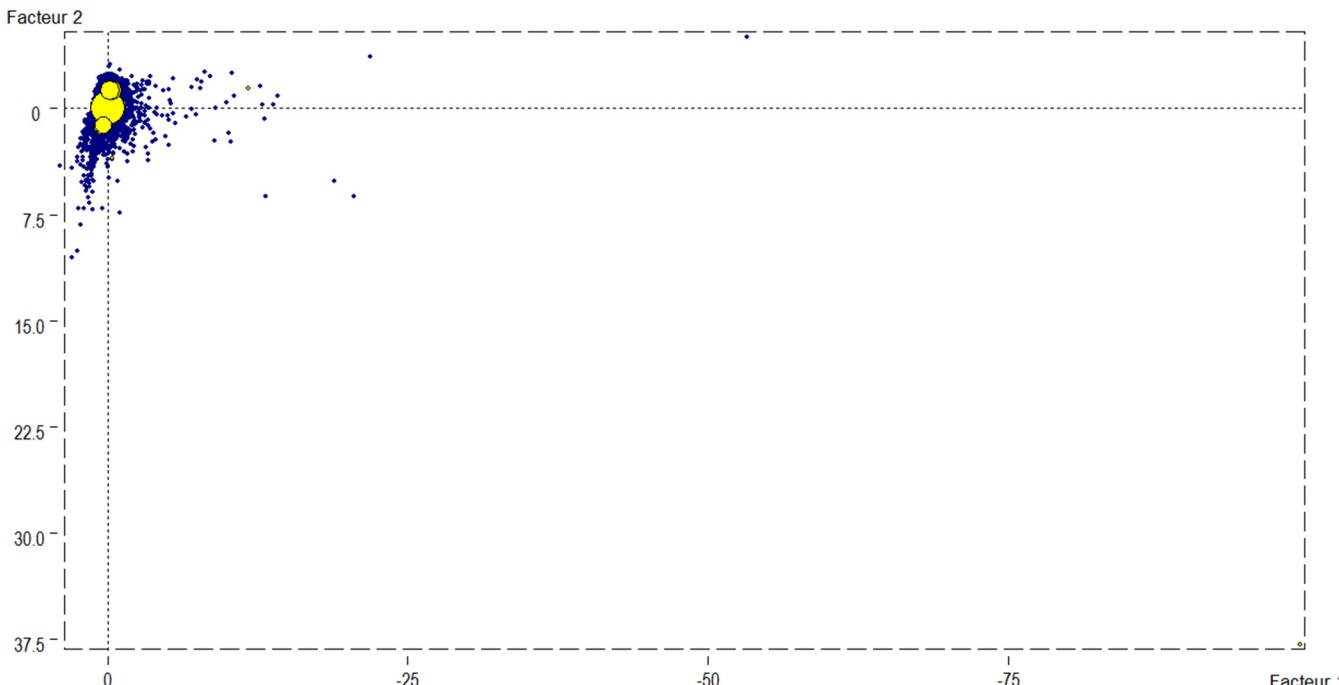

**Figure A4.** Cluster analysis dendrogram and graph for the southern geographic distribution.

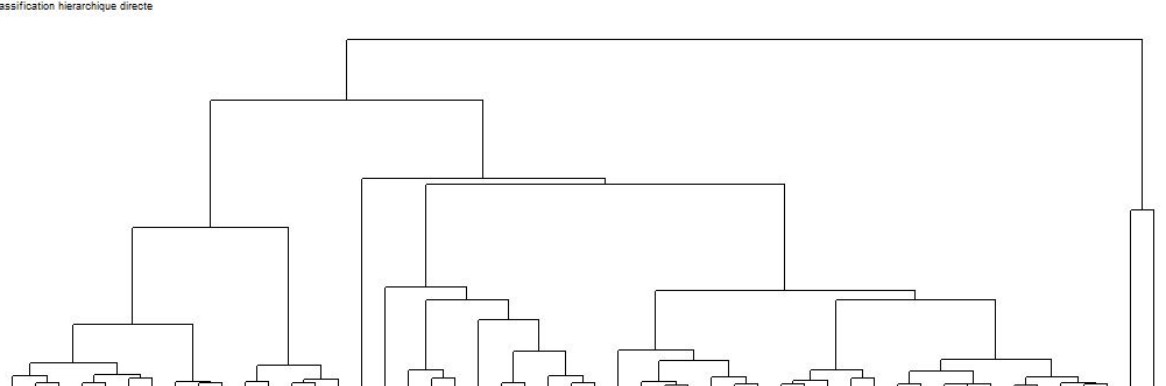

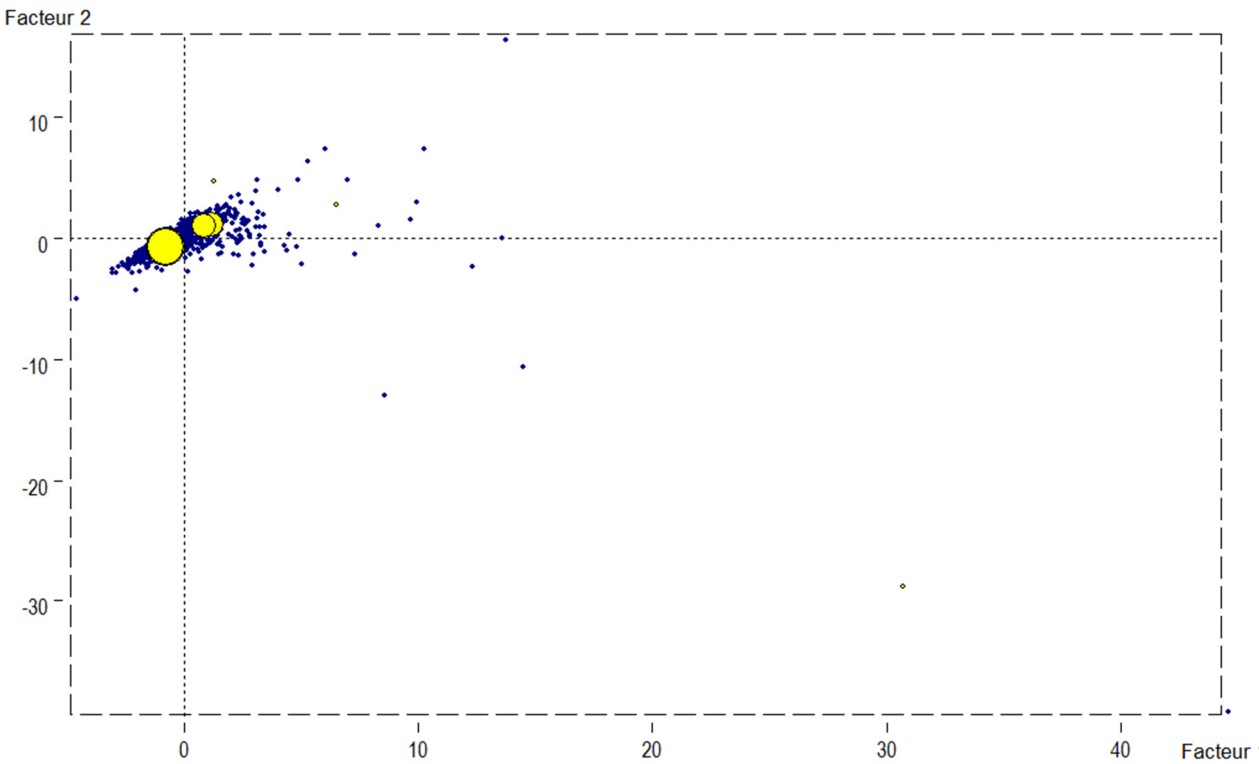

**Figure A5.** Cluster analysis dendrogram and graph for the islands geographic distribution.

## Appendix B

**Table A1.** Description of the average values of the main structural variables and of the main indexes by cluster identified in the northwest geographical breakdown (2020).

|  | Sample Mean | Homologated Family Farms | Resilience | Large Capitalized Farms | Short Supply Chain and Agro-Food Processing |
|---|---|---|---|---|---|
| TAA | 46.6 | 53.0 | 0.8 | 42.2 | 21.8 |
| UAA | 38.2 | 35.5 | 0.6 | 40.3 | 2.4 |
| ALU | 157.9 | 56.1 | 64.3 | 278.9 | 22.3 |
| GSP | 167,167.6 | 109,137.4 | 120,090.3 | 210,129.0 | 49,289.5 |
| Farm Net Income | 61,539.3 | 54,768.0 | 43,848.0 | 66,632.0 | 45,561.5 |
| Irrigated UAA | 19.1 | 5.0 | 0.5 | 29.5 | — |

**Table A1.** *Cont.*

|  | Sample Mean | Homologated Family Farms | Resilience | Large Capitalized Farms | Short Supply Chain and Agro-Food Processing |
|---|---|---|---|---|---|
| AWU | 1.9 | 2.0 | 1.9 | 1.9 | 2.4 |
| VA | 89,359.4 | 72,269.6 | 62,956.7 | 102,091.4 | 52,832.5 |
| EU Subsidies | 15,732.4 | 9478.0 | 2000.0 | 20,388.1 | 600.0 |
| UAA rate | 0.9 | 0.9 | 0.8 | 0.9 | 0.5 |
| % Family work | 0.9 | 0.9 | 0.9 | 0.9 | 2.0 |
| EU Subsidies rate | 0.1 | 0.1 | 0.0 | 0.1 | 0.0 |
| Land mechanization | 29.0 | 28.7 | 160.4 | 20.9 | 4517.9 |
| GSP processing rate | 0.1 | 0.2 |  | 0.0 | 0 |
| GSP quality rate | 0.0 | 0.0 |  | 0.0 | — |
| GSP direct sales rate | 0.1 | 0.1 | 0.4 | 0.0 | 35.7 |
| Irrigation system rate | 0.6 | 0.3 | 0.7 | 0.7 | — |
| Land net profitability | 7188.2 | 5933.8 | 84,671.4 | 5723.0 | 1,234,171.3 |

**Table A2.** Description of the average values of the main structural variables and of the main indexes by cluster identified in the northeast geographical breakdown (2020).

|  | Sample Mean | Homologated Family Farms | Services Farms | Large Capitalized Farms | Short Supply Chain and Agro-Food Processing |
|---|---|---|---|---|---|
| TAA | 35.0 | 11.7 | 5.1 | 45.2 | 48.1 |
| UAA | 28.3 | 10.1 | 2.4 | 39.5 | 11.9 |
| ALU | 261.6 | 391.3 | 1846.0 | 237.5 | 60.3 |
| GSP | 187,475.5 | 146,071.4 | 288,077.3 | 217,369.0 | 93,827.6 |
| Farm Net Income | 75,668.5 | 77,124.4 | 311,550.3 | 76,012.3 | 39,249.7 |
| Irrigated UAA | 11.2 | 6.8 | 0.8 | 14.5 | 2.7 |
| AWU | 2.1 | 2.3 | 2.9 | 2.0 | 1.7 |
| VA | 110,098.7 | 105,643.8 | 359,623.6 | 115,162.8 | 56,656.1 |
| EU Subsidies | 11,462.0 | 3469.5 | 598.0 | 16,317.8 | 4594.3 |
| UAA rate | 0.8 | 0.9 | 0.5 | 0.9 | 0.3 |
| % Family work | 0.9 | 0.8 | 0.8 | 0.9 | 0.9 |
| EU Subsidies rate | 0.1 | 0.0 | 0.1 | 0.1 | 0.1 |
| Land mechanization | 20.7 | 29.5 | 162.2 | 13.8 | 23.4 |
| GSP processing rate | 0.1 | 0.1 | 0.0 | 0.0 | 0.1 |
| GSP quality rate | 0.1 | 0.1 | 0.0 | 0.0 | 0.1 |
| GSP direct sales rate | 0.1 | 0.1 | 0.1 | 0.1 | 0.1 |
| Irrigation system rate | 0.4 | 0.7 | 0.4 | 0.3 | 0.3 |
| Land net profitability | 6257.2 | 9367.0 | 144,070.8 | 2826.6 | 5204.4 |

**Table A3.** Description of the average values of the main structural variables and of the main indexes by cluster identified in the center geographical breakdown (2020).

|  | Sample Mean | Homologated Family Farms | Short Supply Chain and Agro-Food Processing | Large Capitalized Farms | Farm with Quality Label |
|---|---|---|---|---|---|
| TAA | 46.2 | 18.2 | 86.1 | 37.5 | 117.0 |
| UAA | 38.0 | 15.3 | 25.4 | 32.3 | 102.3 |
| ALU | 92.5 | 38.6 | 110.1 | 55.7 | 333.6 |
| GSP | 116,393.8 | 65,228.0 | 58,607.8 | 60,573.4 | 475,846.9 |
| Farm Net Income | 48,033.9 | 28,901.7 | 38,722.5 | 26,051.0 | 186,115.8 |
| Irrigated UAA | 3.5 | 1.8 | 1.0 | 1.2 | 18.0 |
| AWU | 1.8 | 1.7 | 1.8 | 1.4 | 4.3 |
| VA | 72,862.1 | 50,549.3 | 55,735.2 | 39,024.2 | 275,776.7 |
| EU Subsidies | 12,803.3 | 4066.6 | 7698.6 | 10,174.2 | 39,719.4 |
| UAA rate | 0.9 | 0.9 | 0.3 | 0.9 | 0.9 |
| % Family work | 0.9 | 0.8 | 0.8 | 0.9 | 0.5 |
| EU Subsidies rate | 0.0 | 0.1 | 0.2 | 0.0 | 0.1 |
| Land mechanization | 14.5 | 14.9 | 16.0 | 12.2 | 23.9 |
| GSP processing rate | 0.2 | 0.4 | 0.2 | 0.1 | 0.1 |
| GSP quality rate | 0.0 | 0.1 | 0.0 | 0.0 | 0.0 |
| GSP direct sales rate | 0.1 | 0.2 | 0.2 | 0.1 | 0.1 |
| Irrigation system rate | 0.2 | 0.2 | 0.1 | 0.1 | 0.4 |
| Land net profitability | 3489.4 | 2827.4 | 8497.0 | 1782.3 | 10,600.2 |

**Table A4.** Description of the average values of the main structural variables and of the main indexes by cluster identified in the southern geographical breakdown (2020).

| | Sample Mean | Livestock Farm | Resilience | Homologated Family Farms | Intensive Farm |
|---|---|---|---|---|---|
| TAA | 29.5 | 15.2 | 2.9 | 38.8 | 116.8 |
| UAA | 27.2 | 14.2 | 2.5 | 36.3 | 49.8 |
| ALU | 72.6 | 122.9 | 5.0 | 68.5 | 75.4 |
| GSP | 89,047.9 | 75,817.3 | 44,885.0 | 97,673.6 | 93,694.1 |
| Farm Net Income | 40,114.6 | 35,360.4 | 7481.3 | 43,625.1 | 47,397.3 |
| Irrigated UAA | 4.6 | 4.6 | 1.3 | 4.6 | 0.0 |
| AWU | 1.9 | 2.0 | 1.2 | 1.8 | 1.7 |
| VA | 60,060.7 | 56,714.2 | 20,916.7 | 62,658.0 | 55,021.1 |
| EU Subsidies | 9866.3 | 7960.8 | 944.9 | 11,240.6 | 14,702.8 |
| UAA rate | 0.9 | 0.9 | 0.8 | 0.9 | 0.4 |
| % Family work | 0.8 | 0.7 | 0.8 | 0.8 | 0.8 |
| EU Subsidies rate | 0.2 | 0.2 | 0.0 | 0.2 | 0.2 |
| Land mechanization | 10.7 | 11.7 | 64.6 | 9.1 | 11.2 |
| GSP processing rate | 0.1 | 0.2 | 0.0 | 0.1 | 0.1 |
| GSP quality rate | 0.0 | 0.0 | 0.0 | 0.0 | 0.0 |
| GSP direct sales rate | 0.1 | 0.1 | 0.2 | 0.1 | 0.1 |
| Irrigation system rate | 0.2 | 0.3 | 0.7 | 0.2 | 0.0 |
| Land net profitability | 2414.7 | 2945.7 | 4806.8 | 1896.7 | 1984.8 |

**Table A5.** Description of the average values of the main structural variables and of the main indexes by cluster identified in the islands geographical breakdown (2020).

| | Sample Mean | Homologated Family Farms | Services Farms | Large Capitalized Farms | Short Supply Chain and Agro-Food Processing |
|---|---|---|---|---|---|
| TAA | 35.0 | 11.7 | 5.1 | 45.2 | 48.1 |
| UAA | 28.3 | 10.1 | 2.4 | 39.5 | 11.9 |
| ALU | 261.6 | 391.3 | 1846.0 | 237.5 | 60.3 |
| GSP | 187,475.5 | 146,071.4 | 288,077.3 | 217,369.0 | 93,827.6 |
| Net income | 75,668.5 | 77,124.4 | 311,550.3 | 76,012.3 | 39,249.7 |
| UAA | 11.2 | 6.8 | 0.8 | 14.5 | 2.7 |
| AWU | 2.1 | 2.3 | 2.9 | 2.0 | 1.7 |
| VA | 110,098.7 | 105,643.8 | 359,623.6 | 115,162.8 | 56,656.1 |
| Sub. EU | 11,462.0 | 3469.5 | 598.0 | 16,317.8 | 4594.3 |
| UAA | 0.8 | 0.9 | 0.5 | 0.9 | 0.3 |
| % Family work | 0.9 | 0.8 | 0.8 | 0.9 | 0.9 |
| Sub EU rate | 0.1 | 0.0 | 0.1 | 0.1 | 0.1 |
| Land mechaniz. | 20.7 | 29.5 | 162.2 | 13.8 | 23.4 |
| GSP proces. | 0.1 | 0.1 | 0.0 | 0.0 | 0.1 |
| GSP qual. rate | 0.1 | 0.1 | 0.0 | 0.0 | 0.1 |
| GSP | 0.1 | 0.1 | 0.1 | 0.1 | 0.1 |
| Irrigation system | 0.4 | 0.7 | 0.4 | 0.3 | 0.3 |
| Land | 6257.2 | 9367.0 | 144,070.8 | 2826.6 | 5204.4 |

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
