# Peer review of "The Farm’s Orientation towards Sustainability: An Assessment Using FADN Data in Italy"

_land, doi:10.3390/land12020301_

Round 1

Reviewer 1 Report

Dear Authors,

The manuscript presents a quite interesting study, but for me some aspects  must be improved:

1.     The main aim of the research must be defined!!

2.     The steps of the research in the methodology part must be mentioned? Did you used a face to face interview? Or an on-line one?

3.     In the results part maybe an interpretation of the results would be appropriate….it is much easier for the reader to understand the results!!

4.     I don’t see the discussion part in the paper. Maybe the proposals form you regarding the possible development strategies would be appropiate here!

5.     In the conclusion part I don’t see what authors propose for the future???...anyway the conclusion part is a little bit too short!

Reviewer 2 Report

The purpose of this article should be stated more clearly in the summary.

Bibliographic references must be arranged according to the editorial requirements of MDPI journals.

Reviewer 3 Report

1) The abstract should state more clearly the purpose of the paper and major conclusions. The novelty of the work must be clarified and stated, to increase readership.

2) In lines 54-56 the authors state that:

…this article aims to provide a description of the productive fabric in which Italian farms operate, evaluating the effectiveness of the 2014-2020 programming in terms of sustainability and competitiveness.

However, this objective is not sufficiently pursued, the potential effectiveness of the programs implemented in 2014-2020 in terms of sustainability and competitiveness needs to be further explained in the results and conclusions.

3) Lines180-191:

The data may be better presented in a different way, perhaps in a table format.

4) The methodology followed is that of a factor analysis and although it is well known and accepted, it should be described in this section. There is a lack of explanations regarding the adopted methodology.

5) In the results section there should be information regarding, the contribution of variables to the main factors in the PCA, the Agglomeration schedule, a dendrogram.

Reviewer 4 Report

Reconsider after major revision (control missing in some experiments)

Round 2

Reviewer 1 Report

Well done

Author Response

Dear Review,
We are very happy that you appreciated our efforts and thank you for your valuable comments.
The Authors

Reviewer 3 Report

The authors have taken the proposed recommendations into account, however, the issue of language use in the document remains and requires some editing by a native speaker before publication.

Author Response

Dear Reviewer,

thank you again for your valuable comments, as you suggested, we subjected the article to a second reading by a native English-speaking colleague and consequently we further made some revisions on the manuscript. We hope that now the English quality of our article is further improved.

Best regards

The Authors